# Gold nanocarriers for transport of oligonucleotides across brain endothelial cells

**Nayab Fatima[1], Radka Gromnicova[1], Jane Loughlin[1], Basil Sharrack[2], David Male[1]***

**1** Department of Life, Health and Chemical Sciences, The Open University, Milton Keynes, United Kingdom,
**2** Academic Department of Neuroscience and Sheffield, NIHR Translational Neuroscience BRC, Sheffield Teaching Hospitals, NHS Foundation Trust, University of Sheffield, Sheffield, United Kingdom

* David.Male@open.ac.uk

## Abstract

Treatment of diseases that affect the CNS by gene therapy requires delivery of oligonucleotides to target cells within the brain. As the blood brain barrier prevents movement of large biomolecules, current approaches involve direct injection of the oligonucleotides, which is invasive and may have only a localised effect. The aim of this study was to investigate the potential of 2 nm galactose-coated gold nanoparticles (NP-Gal) as a delivery system of oligonucleotides across brain endothelium. DNA oligonucleotides of different types were attached to NP-Gal by the place exchange reaction and were characterised by EMSA (electrophoretic mobility shift assay). Several nanoparticle formulations were created, with single- or double-stranded (20nt or 40nt) DNA oligonucleotides, or with different amounts of DNA attached to the carriers. These nanocarriers were applied to transwell cultures of human brain endothelium in vitro (hCMEC/D3 cell-line) or to a 3D-hydrogel model of the blood-brain barrier including astrocytes. Transfer rates were measured by quantitative electron microscopy for the nanoparticles and qPCR for DNA. Despite the increase in nanoparticle size caused by attachment of oligonucleotides to the NP-Gal carrier, the rates of endocytosis and transcytosis of nanoparticles were both considerably increased when they carried an oligonucleotide cargo. Carriers with 40nt dsDNA were most efficient, accumulating in vesicles, in the cytosol and beneath the basal membrane of the endothelium. The oligonucleotide cargo remained attached to the nanocarriers during transcytosis and the transport rate across the endothelial cells was increased at least 50fold compared with free DNA. The nanoparticles entered the extracellular matrix and were taken up by the astrocytes in biologically functional amounts. Attachment of DNA confers a strong negative charge to the nanoparticles which may explain the enhanced binding to the endothelium and transcytosis by both vesicular transport and the transmembrane/cytosol pathway. These gold nanoparticles have the potential to transport therapeutic amounts of nucleic acids into the CNS.

## Introduction

In recent years, there has been great interest in the potential for gene-therapy or gene editing to treat conditions caused by single-gene defects. Many of these conditions affect the central

**Data Availability Statement:** All relevant data are within the manuscript and its Supporting Information files.

**Funding:** NF was supported by a post-graduate studentship from the Open University and funding from Sheffield Hospital Trust. The funders had no

role in study design, data collection and analysis, decision to publish or preparation of the manuscript.

**Competing interests:** The authors declared that no competing interests exist.

nervous system; examples include Huntington's disease, hereditary forms of Alzheimer's disease, lysosomal storage diseases and spinal muscular atrophy. One approach used to treat Huntington's disease is the use of microRNAs to suppress production of toxic proteins [1], while another technique is to excise the defective gene segment with endonucleases [2]. Other neurological conditions have been treated with anti-sense oligonucleotides [3] or viral vectors carrying a gene to replace the defective host gene [4].

A major obstacle in treating diseases of the CNS is the blood-brain barrier [5], which effectively excludes all hydrophilic biomolecules greater than 1kDa. Large biomolecules including nucleic acids and cytokines are unable to move through the tight junctions between brain endothelial cells [6]. Consequently, it has been necessary to deliver gene- therapy by invasive methods, such as direct intraventricular or intrathecal injection [7, 8]. With these approaches, the distribution of the therapeutic agent is limited to the site of injection or areas close to the ventricles. In contrast, transport of the agent across the brain endothelial cells of the blood brain barrier following less invasive intravascular injection would potentially provide a broader and more even distribution of the treatment. The overall aim of our research is to develop methods for transporting large therapeutic biomolecules into the brain. The purpose of this study was to determine the conditions that affect the transport of oligonucleotides across the blood brain barrier by gold nanocarriers.

A variety of nanoparticles have been used to transport therapeutic agents across the blood-brain barrier [9], including gold nanoparticles [10, 11]. Previously, we have shown that small gold nanoparticles (2nm) are able to rapidly cross brain endothelium in vitro and in vivo and enter astrocytes or other cells of the brain [12, 13]. The small size of these nanoparticles means that they are able to cross brain endothelium by vesicular transcytosis or by direct movement across the apical and basal plasma membranes via the cytosol [14]. Another advantage of these nanoparticles is their relative safety—they have already been used in phase 1 and phase 2 clinical trials for delivery of insulin. The FDA approval process for these trials included 24hr and 28day toxicity and pathology studies in rats and pigs as well as genotoxicity in bacteria. The nanoparticles are synthesised with a covalently-bound coat of sugar molecules which maintains their solubility and limits removal by the mononuclear phagocyte system in vivo. The sugars (glucose or galactose) are attached during synthesis of the nanoparticles by a disulphide bond, which lends itself to an exchange reaction; the bound sugar can be exchanged subsequently with other biomolecules (cargo) that have a free—SH group. The conjugates are stable for months when stored in an oxidising environment but will progressively release their cargo in a reducing environment [15]. In other studies, gold-RNA or -DNA conjugates have been used to modulate gene expression in cells outside the CNS [16], indicating that the oligonucleotides retain their biological activity after internalisation by target cells.

In the current study oligonucleotides have been attached to the gold nanoparticles, to investigate the potential for their transport across brain endothelium and uptake by astrocytes in vitro. We have used the brain endothelial cell line hCMEC/D3 in transwell assays and a 3-dimensional hydrogel astrocyte co-culture model of the blood-brain barrier to investigate the uptake and transport of gold nanoparticles with bound oligonucleotide cargos. Specifically, we have investigated the transport of nanoparticles carrying varying amounts of single or double-stranded oligonucleotides of 20 or 40 nucleotides in length.

DNA oligonucleotides of 20–40 nucleotides were chosen as cargo as DNA is less susceptible to degradation in vitro than unmodified RNA and the length is in the order of that typically employed for therapeutic nucleic acids. The intracellular route taken by nanoparticles has been examined using electron microscopy and the distribution of the oligonucleotide cargo has been assessed by qPCR.

The results show that this class of gold nanoparticle can rapidly and effectively transport oligonucleotides across brain endothelial cells and into astrocytes, in quantities that could target gene-editing or modulate protein expression. Although this model system was designed to investigate oligonucleotide transport, rather than produce a treatment for any one disease, the system should be generally applicable for transport of oligonucleotides across the blood-brain barrier.

## Materials and methods

### Gold glyconanoparticles

The gold nanoparticles (NP), obtained from Midatech Pharma Plc, were synthesised using a modified Brust-Schiffrin method [17] and characterised in water using DLS (dynamic light scattering) on a Nano ZSP Zetasizer (Malvern instruments). They were capped during synthesis with thiol-C2-galactose. The gold core of the nanoparticles was 2nm in diameter, as determined by transmission electron microscopy, and the size of the glyconanoparticles was $3.63 \pm 1.09$ nm (mean $\pm$ SD) measured by DLS. As determined by molecular mass of the gold core (~20,000) and physical size, these nanoparticles have approximately 100 gold atoms in the core and 40 covalently bound C2-galactose molecules on the outside, attached by disulphide bonds.

Gold in the NP formulations was measured by a spectrophotometric method against gold standards (Sigma). The assay was performed in a 96 well plate, with a total volume of 200µl in each well. In each well, 10µl of sample, 10µl $H_2O$ and 30µl of 100% freshly prepared aqua regia (kept on ice) were added. The liquids were then mixed by gentle tapping and left to incubate for 1 min. Next, 150µl of 2M NaBr was added. The absorbance was read on a plate reader using OPTIMA FluoSTAR, at 390 nm. The concentrations (µg/ml) of NPs used in the biological assays refers to the gold concentration.

### Oligonucleotides and attachment to nanoparticles

A 20nt, thiolated ssDNA oligonucleotide (5' Thiol-C6- AAT ATC GCG GAC AGA AGA CG 3') was obtained from Sigma. This sequence was derived from a plasmid of *Neisseria gonorrhoeae* (pCmGFP) (GenBank: FJ172221.1) and was chosen on the basis that it has no homology in the human, rat or mouse genome and could therefore be readily detected by qPCR in human, rat or mouse systems without interference from genomic DNA. The oligonucleotide as supplied was substantially oxidised (at the 5' thiol), and hence was dimerised. Before attachment to the glyconanoparticles, Tris(2-carboxyethyl)phosphine (TCEP, in 0.1M tris/HCl, pH 7.5) was used to reduce the DNA to yield free thiol monomers. This was achieved by incubating the oligonucleotide (7.5mM in 50µl) at room temperature for 4 hours with TCEP at a molar ratio of 1: 1.25 (DNA:TCEP).

**NP-DNA-20ss.** After reduction, the oligonucleotide was attached to the NPs by an exchange reaction. NPs (2mg/ml equivalent of gold) were mixed with reduced 20nt ssDNA at a molar ratio of 1:14 in a 0.5ml Eppendorf tube, in a sealed container flushed with nitrogen gas to exclude oxygen. The reaction was allowed to continue for 48 hours at room temperature unless otherwise stated.

**NP-DNA-20ds.** In order to make nanoparticles with 20nt double-stranded DNA attached, the NP-DNA-20 preparation was reacted at room temperature with an equimolar amount of complementary single-stranded DNA (5'-CGTCATCAGTCCGCGATATT -3').

**NP-DNA-40.** To synthesise nanoparticles with 40nt double-stranded DNA, the NP-DNA-20ss preparation was first hybridised with an equimolar amount of 40nt ssDNA, half complementary to the 20nt ssDNA bound to the NP (5'–

AAAAGCTCTGCCTTGGTTTCCGTCTTCTGTCCGCGATATT-3') for 30 minutes. Then the extended single-strand of the 40nt oligonucleotide was filled in by hybridisation with an equimolar amount of 20nt ssDNA (5'– GAAACCAAGGCAGAGCTTTT– 3'). A diagram showing the arrangement of the different oligonucleotides attached to the gold core is shown in S1 Fig in S1 File.

**FPLC of nanoparticles.** In some experiments the nanoparticles were fractionated by FPLC (fast protein liquid chromatography) to remove any unbound oligonucleotides and to separate NPs with high or low levels of bound oligonucleotides. This technique is normally used to separate proteins, of 10–1000 kDa, but it is equally effective for gold glyconanoparticles [18]. The nanoparticles were fractionated on an ÄKTA pure FPLC system (GE Healthcare) using a Superdex 200 10/300 GL column in phosphate buffered saline (PBS). The elution volume for all fractions was set at 0.5ml at a flow rate of 0.4 ml/min. Detection of oligonucleotides and nanoparticles was by UV absorption. The pooled fractions with different amounts of bound 40nt dsDNA are designated **NP-DNA-40**$^{Hi}$ and **NP-DNA-40**$^{Lo}$.

**EMSA of nanoparticles.** Nanoparticles were analysed by electrophoretic mobility shift assays (EMSA). This technique is normally used to determine whether proteins have bound to segments of DNA, but in this case, it was adapted to analyse oligonucleotide attachment to the nanoparticles. In these experiments the 20nt complementary DNA sequences were substituted with 5'-biotinylated versions, to allow detection of oligonucleotides on nanoparticles on the EMSA blots (Bio-CGTCATCAGTCCGCGATATT or Bio-GAAACCAAGGCAGAGCTTTT). Complementary biotinylated DNA was hybridised with NP-DNA-20$^{ss}$ to form NP-DNA-20$^{ds}$. A molar ratio of 1nmol of NP-DNA-20$^{ss}$ to 1.15 nmol of 20nt biotinylated ssDNA (complementary to thiol-C6-DNA) was used. The sample mixture was adjusted to pH ~ 8 by adding 10X TBE buffer, leaving the final concentration of the TBE buffer 0.5X (pH 8.3) in the sample mixture. The mixture was incubated for 30 min at room temperature for the hybridization to take place. The EMSA technique is highly sensitive, so the samples were diluted x100 in pure water and then prepared with 1 µl NP sample, 6 µl 5X TBE, 13 µl water and 5 µl loading dye (6X, Fermentas).

NP-DNA-20$^{ds}$-biotin were separated on a 6% polyacrylamide gel and NP-DNA-40$^{ds}$-biotin on a 5% gel. The gels were pre-run in 0.5x TBE at 85 V for 1hr. The samples (20µl in each well) were then loaded onto the gel and run at 90V for 70 mins for NP-DNA-20$^{ds}$-biotin and 80 mins for NP-DNA-40$^{ds}$-biotin. The gels were transferred onto nylon membrane (Amersham Hybond N+: GE Healthcare) at 100 V for 1hr and 15 mins in 0.5x TBE at 4° C. DNA was then cross-linked onto the membrane with UV light (120mJ/cm$^2$) and developed using a chemiluminescent DNA detection kit, according to the manufacturer's instructions (Thermofisher) and the blots imaged with a GelDoc.

## Tissue culture systems

The human brain endothelial cell line hCMEC/D3 originally characterised in this laboratory [19] was used at passage 26–30, cultured in modified EBM-2 (Lonza)—containing 0.025% VEGF, IGF and EGF; 0.1% bFGF, 0.1% (v/v) rhFGF, 0.1% (v/v) gentamycin, 0.1% (v/v) ascorbic acid, 0.04% (v/v) hydrocortisone. The amount of serum was reduced to 2.5% (v/v) of foetal bovine serum. Cytotoxicity of gold nanoparticles on these cells was measured by an Alamar blue assay [20].

Primary human astrocytes (passage 2–7) obtained from ScienCell were cultured in astrocyte medium (ScienCell)–supplemented according to manufacturer's instructions with FBS (2%), astrocyte growth supplement (1%), and penicillin/streptomycin solution (1%).

Transwell cultures for transport assays were produced by seeding hCMEC/D3 cells onto collagen-coated 0.4 μm polyester membrane transwell inserts (Corning Costar). The transwell inserts used were either 6.5 mm (seeding density 20,000 cells/insert) or 12 mm (seeding density 70,000 cells/insert). The hCMEC/D3 cells were grown until confluent (2–3 days). At this point, nanocarriers (8 μg/ml gold) were added to the upper chamber in EBM-2 MV medium and incubated at 37˚C for 3 hours. After the incubation, both upper and lower chambers were washed 3x in 0.5ml Hank's Balanced Salt Solution (HBSS). The cells were fixed for 1 hr at room temperature in 2.5% glutaraldehyde in 0.1M PB (Sörensons phosphate buffer). The fixative was removed and the chambers were washed 3x in PBS and stored in 0.1M PB at 4 ˚C. The inserts were then processed for electron microscopy to localise and quantitate the gold nanoparticles.

For quantitation of DNA (qPCR), the medium was directly harvested from the upper and lower chambers, and the cells were lysed in 1% Triton X100. These samples were then immediately frozen at -20˚C, until assayed.

A 3-dimensional co-culture of human brain endothelial cells (hCMEC/D3) and human astrocytes (hA) was used for nanoparticle uptake and transport assays [21]. Astrocytes at 1.5 x10$^6$/ml were seeded into a collagen gel solution, prepared using 80% (v/v) rat type I collagen (2 mg/ml), 10% 10x MEM and 10% human astrocyte cell suspension in astrocyte medium (500μl/well). The gelling of the collagen+MEM was initiated by careful, dropwise addition of 5M NaOH, to neutral pH, before addition of the astrocytes. The gel was incubated for 20 min at 37 ˚C then RAFT™ absorber plungers (Lonza) were dropped carefully on top of the collagen gels to absorb ~90% of the liquid. The RAFT™ absorber plunger was removed after 20 mins. Astrocyte medium was added to the compressed gels, and the culture was incubated for 24hr. hCMEC/D3 cells were then seeded on top of the gel and grown over 2 days in modified EBM-2 medium until they formed a monolayer.

For transport experiments, nanoparticles were added to the 3-dimensional co-culture gels at a final concentration of 8 μg/ml (gold) in modified EBM-2 media and incubated for 3 hrs at 37˚C. After the incubation, the medium was removed, the cultures were washed 3x in 0.5ml HBSS and fixed in 2.5% glutaraldehyde for 1 hr. The compressed gels were washed 3x in PBS and gently detached from the bottom of the 24 well plates with a spatula. They were then stored in phosphate buffer at 4˚C and processed for electron microscopy.

## Localisation and quantitation of nanoparticles by transmission electron microscopy

Cells were permeabilized in 0.01% Triton X100 for 15 minutes on a rocker and then washed 3x in 0.1M PB. Silver enhancement solution (Aurion) was prepared according to the manufacturer's instructions and applied to the fixed transwell inserts for 1 hr and 10 mins on a rocker and then washed 3x in distilled water. Cells were then osmicated in 1% (w/v) osmium tetroxide (in 0.1M PB) for 30 min and then washed 3x in 0.1M PB. The insert was removed from the well, the membrane was cut out and strips (~3 mm wide) were prepared from the centre of the membrane. Finally, the membranes were dehydrated by immersion in increasing concentrations of ethanol as follows: 30% ethanol for 5 min, 50% ethanol for 5 min, 70% ethanol for 10 min, 100% ethanol for 10 min (performed twice), 100% ethanol with molecular sieve for 10 min.

After dehydration, the membrane strips were incubated in a 50:50 mixture of 100% ethanol and Epon resin (Agar) and placed on a rocker overnight. The next day, the insert membrane pieces were penetrated with freshly prepared Epon resin and changed twice, each change was

incubated for 2 hrs. The insert membrane pieces were embedded in Epon resin on a cushion pad for 48 hrs at 60 ˚C.

Microsections of 80nm were collected onto pioloform film-covered copper grids. After at least 2 hrs of air drying, they were counterstained with 3% aqueous uranyl acetate for 30 mins followed by Reynolds lead citrate for 10 mins. Minor modifications of this protocol were needed for preparation of the 3D cultures [21].

The sections were observed at a magnification of 20,000x on JEM 1010 (Jeol). The nanoparticles were counted in cells and sorted into 4 cell compartment categories; vesicles, cytosol, basal plasma membrane (under cell) and nucleus as previously described [12]. The length of the insert which was assessed on TEM ranged usually between 300–1500 microns in length. The number of nanoparticles was then calculated per micron of insert.

## Quantification of DNA by qPCR

SYBR Green 2X master mix kit was from Qiagen. Stock forward and reverse primers, corresponding to the sequence of the 40nt dsDNA (S1 Fig in S1 File) were made up at 10μM. The 20μl reaction mixture, assembled on ice, contained 10μl 2X SYBR green mastermix, 1μl of each of the forward and reverse primers (0.5 μM final concentration), 7μl water and 1μl DNA template. The template was diluted to give 1-10pg DNA in the reaction mix. Amplification and detection conditions were 2 min at 50˚, 10 min at 95˚C, followed by 40 cycles of 15 s at 95˚C, and 1 min at 55˚C. The Ct values for each sample were normalised using a standard curve that was established using the 40nt ssDNA strand used to prepare the NP-DNA-40.

## Statistical methods

Data was analysed and graphs prepared using Graphpad Prism 8.0. Tests used, replication and P values are reported in the figure legends.

## Results

### Synthesis and characterisation of nanoparticles with bound oligonucleotides

It was necessary to first establish the conditions for production of gold glyconanoparticles (NP) with attached oligonucleotides, by the place exchange reaction. Gold glyconanoparticles with a 2nm core and coat of thiol-C2-galactose were mixed with reduced thiol-C6-DNA (20nt, single stranded) for 1–4 days at a molar ratio of 1:14. In these conditions, the oligonucleotide exchanges with theC2-galactose on the surface of the nanoparticles. The reaction products examined by EMSA (Fig 1A) indicate the position of the oligonucleotide; the 6 bands at the top of the gel, correspond to nanoparticles with different numbers of covalently bound oligonucleotides. Previous work indicates that each band corresponds to one additional oligonucleotide bound to the nanoparticle [22], implying that these nanoparticles can carry up to 6 oligonucleotides. The proportion of larger bands increased up to 2 days but did not increase further at day 3 (S2 Fig in S1 File). By day 4 there was a substantial increase in oxidised (dimeric) oligonucleotide, which is unavailable for the exchange reaction. Based on these observations, 2 days was identified as the optimum time for the exchange reaction in the stated conditions and was used in subsequent preparations.

The reaction mixture was also examined by FPLC on a G200 column (Fig 1B). Fractionation completely separated the unconjugated NPs and free oligonucleotides (monomers and dimers) from the NP-DNA. The NP-DNA conjugates appeared as 3 overlapping peaks. Analysis of these fractions by EMSA confirmed that these peaks contained the NP-DNA bands

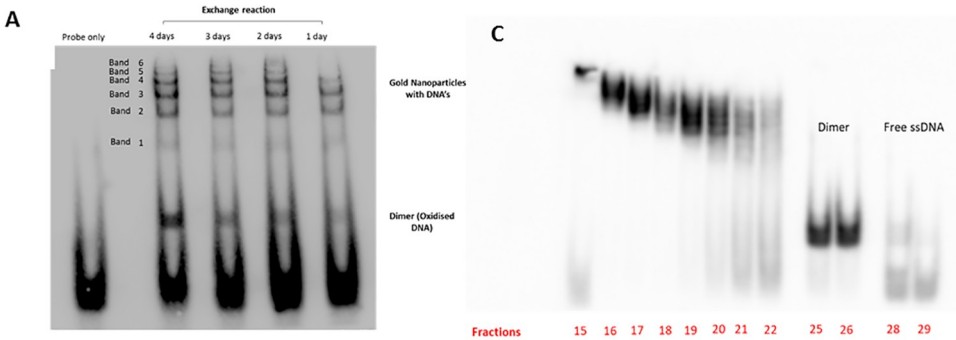

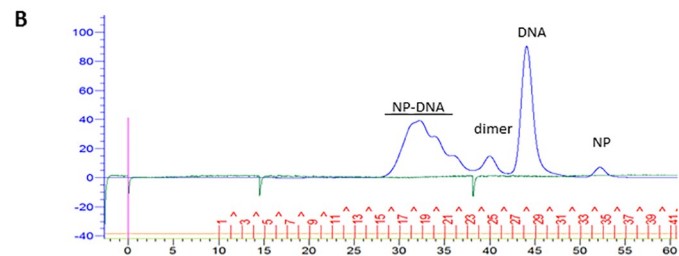

**Fig 1. Production and characterisation of nanoparticles with bound oligonucleotides.** (A) EMSA of exchange reaction (1–4 days) probed with a biotinylated oligonucleotide complementary to the ssDNA oligonucleotide attached to the NPs. (B) FPLC trace of reaction products of exchange reaction with 20nt thiol-C6-ssDNA, at 48hrs. y-axis: pressure and absorption at 280nm. x-axis: time (0–60 minutes) and fraction number (1–41). (C) EMSA of FPLC fractions (15–29) of NP-DNA. The numbers 15–29 correspond to the fraction numbers shown in part Fig1B.

previously identified (Fig 1C). However, FPLC was not able to fully resolve the different NP-DNA conjugates. Therefore, in order to make NP-DNA preparations with different levels of bound oligonucleotide, two pools were prepared: DNA$^{Hi}$ (fractions 15–17); DNA$^{Lo}$ (fractions 19–21). The DNA$^{Hi}$ fraction corresponds to NPs in the upper bands and the DNA$^{Lo}$ fraction with lower bands (Fig 1). These two preparations were subsequently used to prepare NPs with high or low numbers of bound 40nt dsDNA oligonucleotides.

Nanoparticles with different numbers of bound oligonucleotides were examined by transmission electron microscopy (S3 Fig in S1 File), and the size of the gold-cores measured. There was no change in the core size of the NPs with different numbers of oligonucleotides bound, indicating that the attachment of the DNA had not caused aggregation of the NPs. Note that the TEM images identify the core rather than the hydrodynamic diameter of the NPs. However, the FPLC trace does give data on the hydrodynamic diameter of the NP-DNA. In the absence of nanoparticle standards, the G200 column was calibrated using protein standards (S4 Fig in S1 File) and values obtained for each of the FPLC fractions; by interpolation, the mean hydrodynamic diameters of the NPs were: NP = 1.8nm, NP-DNA-20$^{Lo}$ = 3.7nm and NP-DNA-20$^{Hi}$ = 4.9nm. The absolute values for the NPs are unlikely to be highly accurate, as they were measured against protein standards and the nanoparticles with bound DNA are asymmetric. Nevertheless, it indicates that attachment of the 20nt ssDNA oligonucleotides increases the effective hydrodynamic diameter of the NP-conjugates by 2–3 fold.

To prepare NPs with 40nt dsDNA oligonucleotides, the NP-DNA-20$^{Hi}$ or NP-DNA-20$^{Lo}$ preparations were hybridised with 40nt ssDNA, with its 3' segment complementary to the oligonucleotides attached to the nanoparticle. The remainder of the primary strand was hybridised with a 20nt complementary ssDNA, to make 40nt dsDNA (S1 Fig in S1 File). Unreacted,

free oligonucleotides were removed by FPLC, and the products (NP-DNA-40$^{Hi}$ and NP-DNA-40$^{Lo}$) verified by EMSA (S5 Fig in S1 File). The hydrodynamic diameters of the NPs with 40nt dsDNA attached were determined as before (S4 Fig in S1 File): NP-DNA-40$^{Lo}$ = 7.5nm and NP-DNA-40$^{Hi}$ = 7.6nm.

Previous studies have shown that the base NPs were not cytotoxic for hCMEC/D3 cells up to 32μg/ml [12]. The NP-DNA-40$^{Hi}$ and NP-DNA-40$^{Lo}$ preparations were also checked for potential cytotoxicity on these endothelial cells with an Alamar blue assay (S6 Fig in S1 File). There was no detectable cytotoxicity up to 32μg/ml (gold concentration) over 24 hours. In the assays described below, NPs were used in cell cultures at 8 μg/ml for 3 hours.

## Uptake and transport of DNA nanocarriers by brain endothelium

The uptake and transport of nanoparticles with 20nt ssDNA or dsDNA (unfractionated) was compared, using TEM to quantitate nanoparticles in different cellular compartments of hCMEC/D3 cells (Fig 2A). The data is represented as number of nanoparticles per micron of insert membrane the cells grew on. The compartments assessed were: cytosol, vesicles, nucleus and under the basal plasma membrane. Examples of electron micrographs with nanoparticles in different subcellular compartments are shown in Fig 2B. Nanoparticles that have crossed the basal plasma membrane indicating trans-endothelial transport are listed as 'undercell'. Note that the silver enhancement increases the size of the NPs for easier visualisation.

Despite the increase in size of the nanoparticles, the attachment of DNA did not reduce or block their transport. In contrast, nanoparticles with double stranded DNA were present in greater numbers in vesicles than NP-DNA$^{ss}$ or control NP-Gal.

The transport rate of NPs with a 20nt dsDNA cargo or an equivalent 40nt dsDNA cargo (unfractionated) was compared (Fig 3A). Examples of electron micrographs indicating the localisation of nanoparticles are shown in Fig 3B. NP-DNA-40$^{ds}$ were taken up into cytosol and vesicles more efficiently than NP-DNA-20$^{ds}$. Moreover, the rate of transcytosis of NP-DNA-40$^{ds}$ was much greater than NP-DNA-20$^{ds}$ or for NP without any bound DNA, as indicated by the number of NPs that had crossed the basal membrane of the endothelium.

To investigate the effect that different numbers of bound oligonucleotides have on transport, the rate of uptake of fractionated NP-DNA-40$^{Hi}$ and NP-DNA-40$^{Lo}$ were compared (Fig 4). As noted previously, the number of NPs with attached oligonucleotides that had crossed the basal membrane was significantly greater than that of the NPs without cargo, however there was no consistent difference in transcytosis when comparing NPs with a high density of DNA and those with a low density. In interpreting these experiments, it is important to note that the results show a single time point (3hrs) in a dynamic process. Previous work has shown that transcytosis of these NPs is detectable within 30 minutes of application both in vitro and in vivo. The NPs accumulated beneath the basal membrane give a quantitative measure of how many NPs have crossed the cells during the time-course (0-3hrs) but the levels in the cellular compartments only detect what is present in those compartments at the 3hr time-point.

## Transport of DNA across brain endothelium

The experiments described above detected the gold nanocarriers by TEM, but not the DNA. It was therefore important to confirm that the DNA remains attached to the nanocarriers as they cross the endothelium and to quantitate the effectiveness of the NPs as carriers. Initially, NP-DNA bands were cut from the EMSA blots and 1mm$^2$ pieces used as templates in the PCR reaction. The products examined by agar gel electrophoresis demonstrated that DNA bound to the gold NPs could still function as a template for PCR. Following optimisation of the

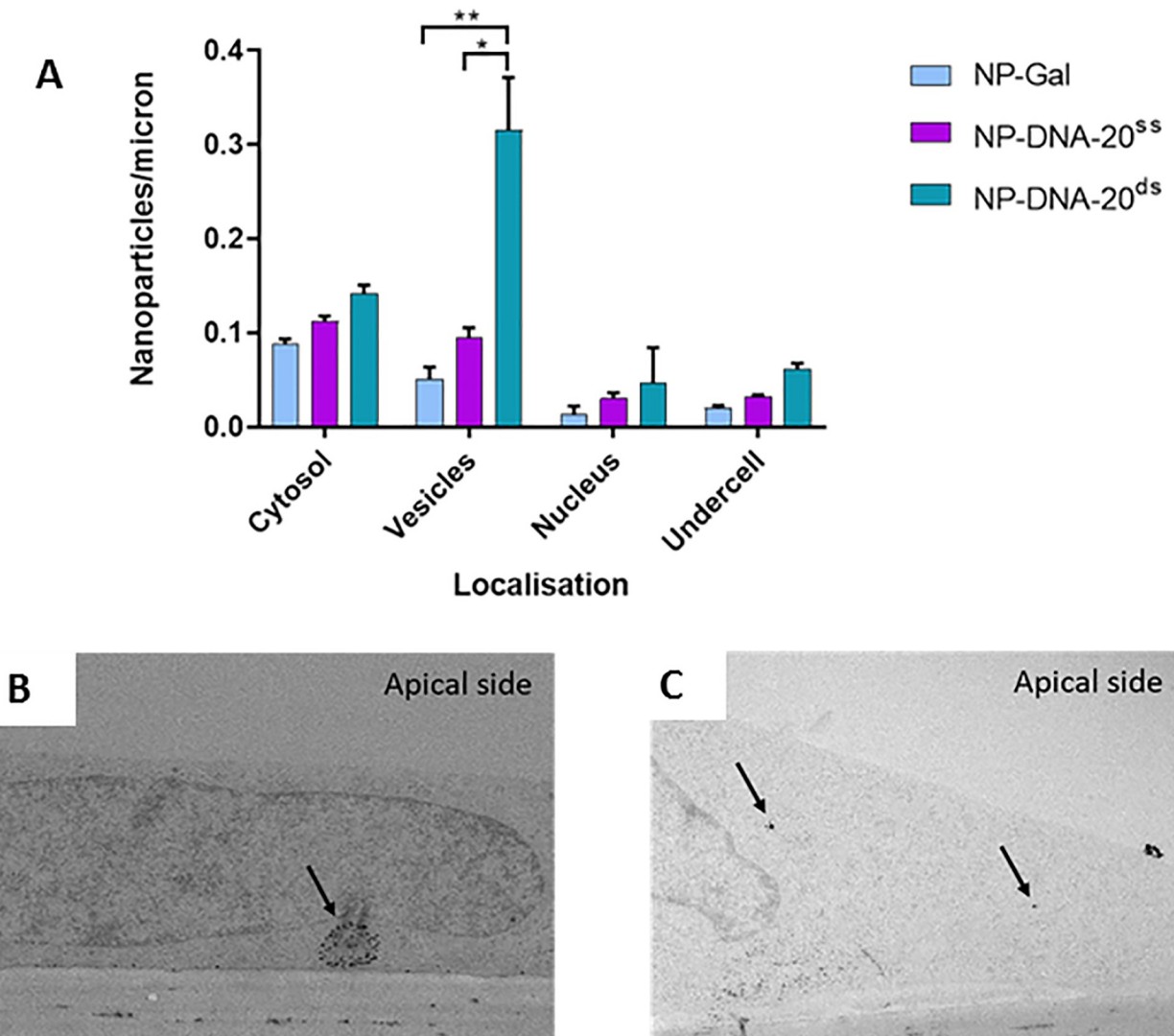

**Fig 2. Endocytosis of gold nanocarriers with ssDNA or dsDNA by brain endothelium.** (A) Nanoparticles in cytosol, vesicles and the nucleus of hCMEC/D3 cells. NPs with 20nt ssDNA (NP-DNA-20ss) were compared with dsDNA(NP-DNA-20ds) and the base nanoparticles (NP-Gal). Three experiments were performed, each individual experiment with three technical repeats. Tukey's multiple comparisons test showed a significant difference for NP-DNA-20ds compared to NP-Gal (** P = 0.0034) and NP-DNA-20ss (* P = 0.0133) in vesicles. (B) Transmission electron micrographs showing examples of silver enhanced gold NPs in vesicles (NP-DNA-20ds, A) or the cytosol (NP-Gal, B) of hCMEC/D3 cells after application to the apical surface.

reaction with different primers, the combination of a 20nt forward primer (`AATATCGCGGA CAGAAGACG`) and 24nt reverse primer with a 4nt overhang (`GAGCAAAAGCTCTGCCTTGGT TTC`) was adopted. This combination allowed the amplicon (44nt) to be distinguished from any residual template (40nt) in the reaction products.

Having established conditions for PCR detection of NP-DNA, the transport of NP-DNA-40Hi, NP-DNA-40Lo and free 40nt dsDNA were compared. Transport of the DNA was measured in hCMEC/D3 transwells collecting medium from the upper and lower chambers and

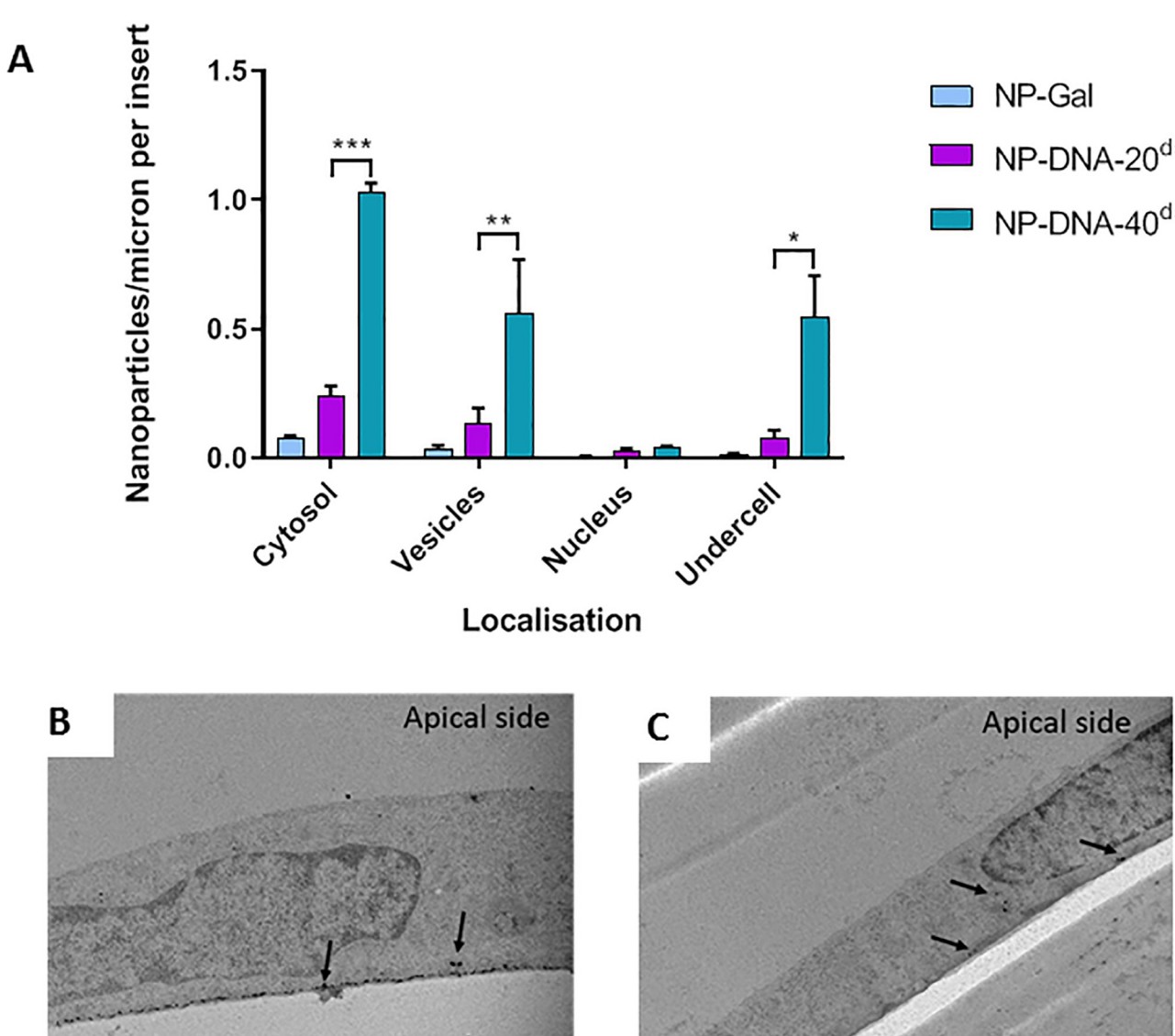

**Fig 3. Endocytosis and transcytosis of nanocarriers with 20nt or 40nt dsDNA cargo.** (A) Uptake of NP-DNA-20[ds] and NP-DNA-40[ds] into brain endothelial cells. Nanoparticles were quantified in the cytosol, vesicles, nucleus, and at the basal membrane (undercell). NP-Gal was the nanoparticle control, no DNA attached. Three experiments were performed, each experiment having three technical repeats. Tukey's multiple comparisons test showed significant difference for NP-DNA-40[ds] compared to NP-DNA-20[ds] in cytosol (*** $P = 0.0003$), vesicles (** $P = 0.0082$) and under cell (* $P = 0.0108$). (B) Electron micrographs of silver enhanced gold NPs in vesicles and at the basal membrane of hCMEC/D3 cells treated at the apical membrane (top) with NP-DNA-40[ds] (A) or NP-DNA-20[ds].

from the endothelial layer following lysis of the cells. Amounts of transported DNA were determined by qPCR and interpolated from a standard curve with known amounts of 40nt ssDNA (Fig 5). The results show that free DNA was not taken up by the cells or transported across the transwells, but significant amounts of DNA were transported to the lower chamber, when attached to the gold nanocarriers. Moreover, NP-DNA-40[Lo] was more efficiently transported into the bottom chamber than NP-DNA-40[Hi].

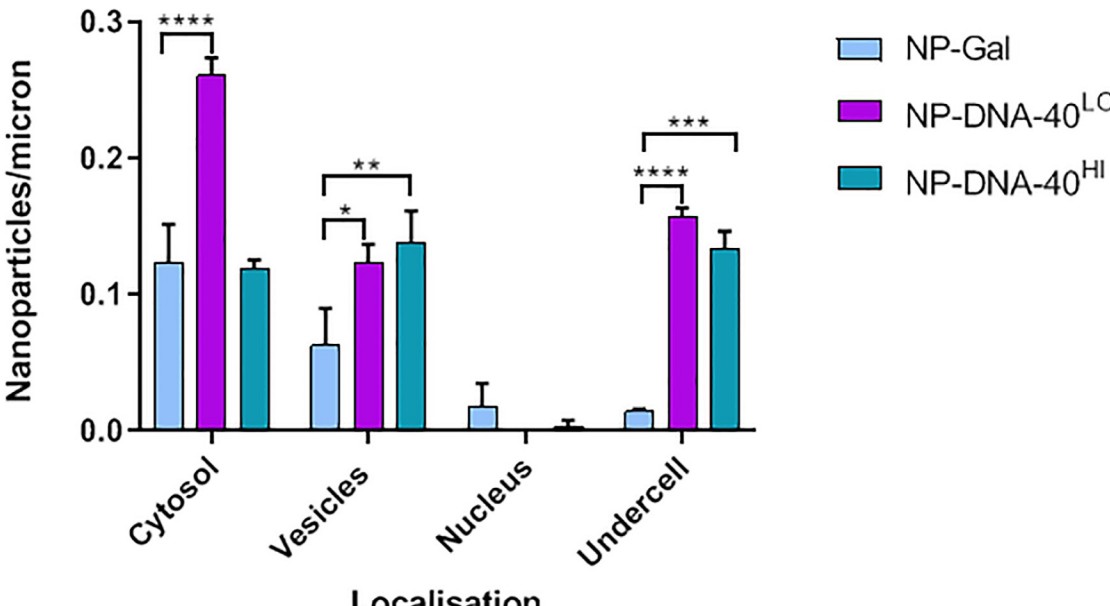

**Fig 4. Endocytosis and transcytosis of nanocarriers with high or low levels of dsDNA cargo.** Nanoparticles with high or low levels of 40nt dsDNA located in cytosol, vesicles and nucleus of hCMEC/D3 cells and beneath the basal membrane (undercell). Three experiments were performed, each individual experiment having three technical repeats. Tukey's multiple comparisons test showed significant difference for NP-DNA-40[Lo] in cytosol, vesicles and "undercell" compared to NP-Gal (**** $P<0.0001$: *** $P<0.001$; ** $P<0.01$, * $P<0.05$).

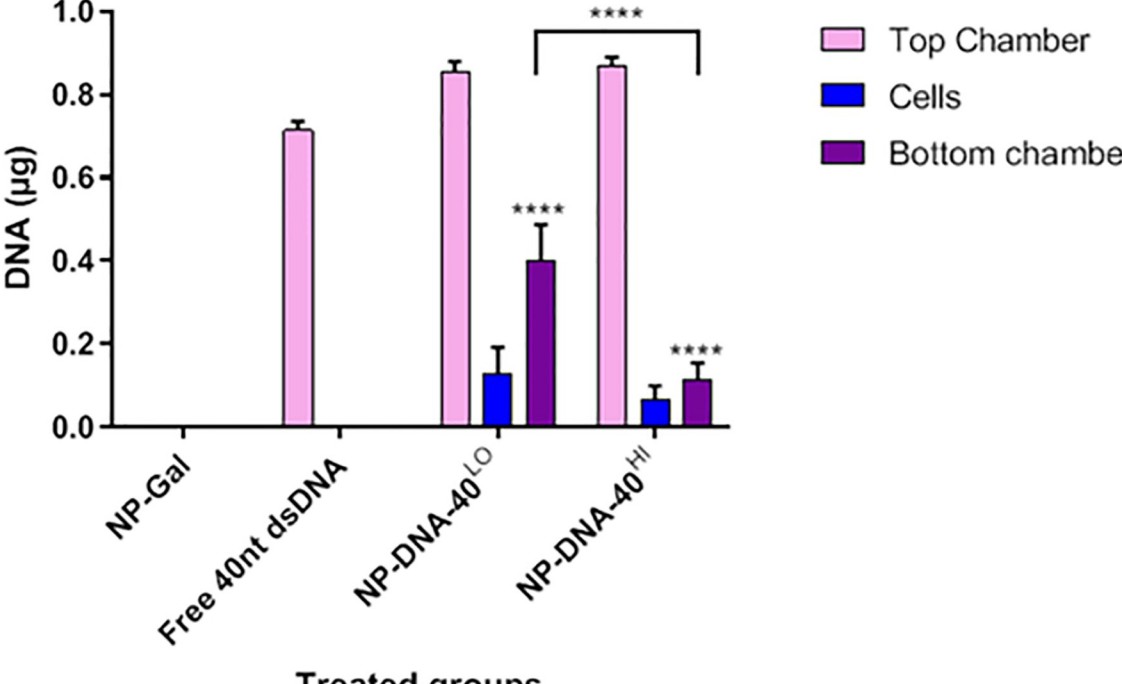

**Fig 5. Transport of DNA across brain endothelium by gold nanocarriers.** qPCR quantitation of 40nt DNA measured in transwells treated with NP-Gal, free 40nt dsDNA, NP-DNA-40[LO] and NP-DNA-40[HI]. Tukey's multiple comparisons test was performed. The level of DNA carried across the endothelium was significantly greater with either of the NP-DNA formulations than for free DNA (**** $P<0.0001$) and transport by NP-DNA-40[Lo] was higher than NP-DNA-40[Hi].

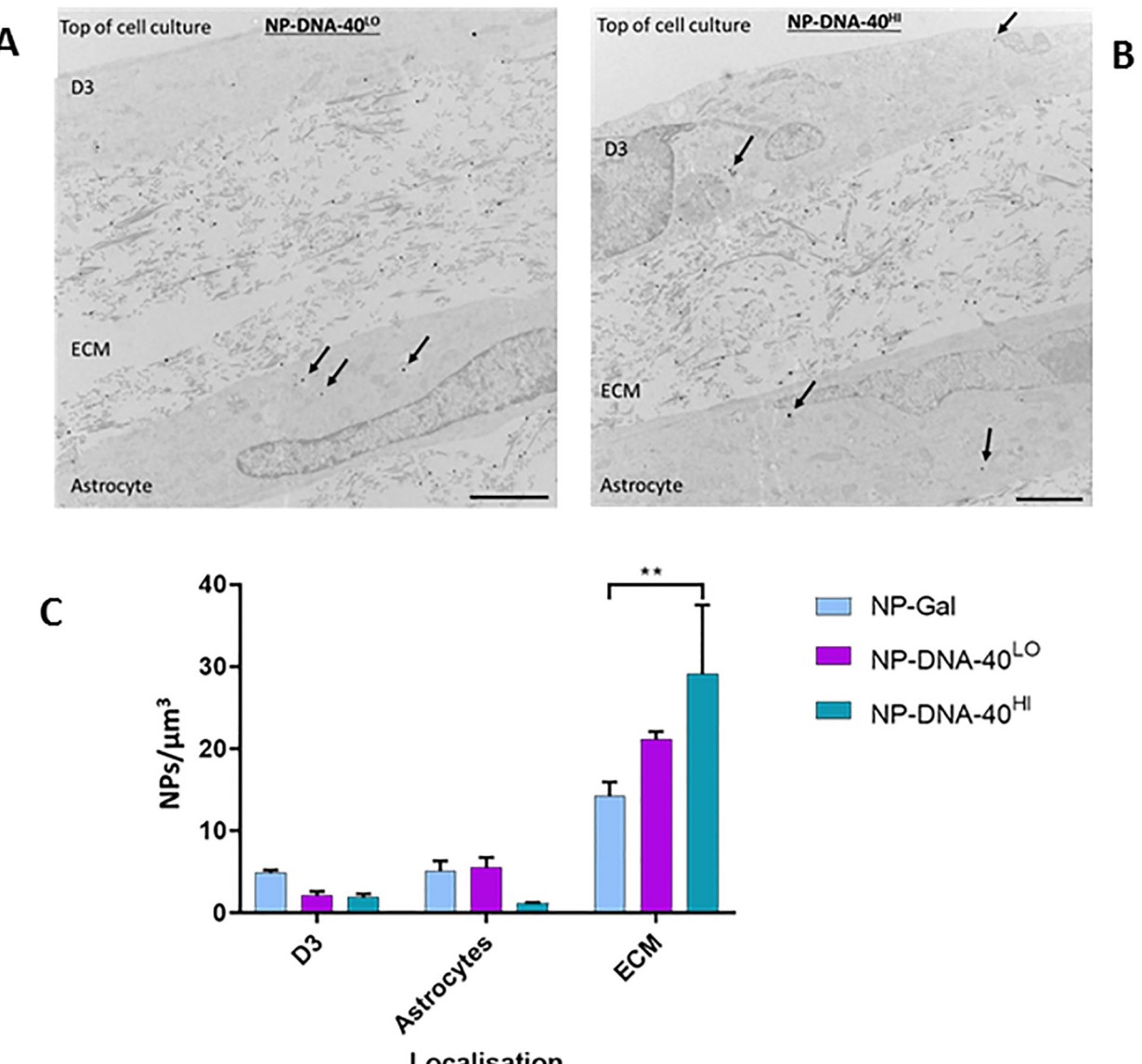

**Fig 6. Endocytosis of nanocarriers with dsDNA cargo in a blood-brain barrier model.** (A and B) Representative TEM images from 3-independent experiments, showing sliver-enhanced hydrogel cocultures of hCMEC/D3 cells (D3) overlying collagen extracellular matrix (ECM) with embedded astrocytes. Arrows indicate examples of nanoparticles. Scale bars = 2.0 μm. (C) Quantitation of nanoparticles in endothelium, astrocytes and collagen ECM. The number of NP-DNA-40$^{Hi}$ in the matrix was significantly greater than for NP-Gal control, by Tukey's multiple comparison test (P<0.01).

## Uptake of nanocarriers by astrocytes

To determine whether these nanocarriers are able to access target cells, uptake was assessed in astrocytes in a 3-dimensional collagen hydrogel model of the blood brain barrier. In this system, the nanocarriers must first cross the endothelium, enter the extracellular space beneath the basal plasma membrane of the endothelium and then be internalised by astrocytes lying within the collagen gel (Fig 6A and 6B). Because the volume of astrocytes, endothelium and collagen matrix in these cultures differs greatly, the results are expressed as number of nanoparticles per μm$^3$, where volume is determined by the total area occupied by the different cells (BECs or astrocytes) or collagen matrix in the TEM images multiplied by thickness of the

sections (80nm). The results show that nanoparticles do enter the astrocytes, although the density of nanoparticles is higher in the matrix than in either of the two cell types (Fig 6C). There was no significant difference between levels of different nanoparticle types in the astrocytes, although unlike endothelial cells in the transwell system, there was a higher proportion of cytosolic uptake than vesicular internalisation in astrocytes (S7 Fig in S1 File). The results demonstrate that these nanoparticles can move across the endothelial cells and be taken up by astrocytes.

## Discussion

The work of our group has shown that gold nanoparticles rapidly cross brain endothelium in vitro and in vivo [13, 23]. Here, we have replaced some of the galactose ligand on the nanoparticles with oligonucleotides to model a therapeutic cargo. The adaptation of FPLC to fractionate gold nanoparticles with different stoichiometries and the development of EMSA for characterisation of nanocarriers has allowed us to produce and characterise gold nanoparticles with different types and amounts of attached oligonucleotides.

Taken together the results show that these gold-NP can transport dsDNA-oligonucleotides across brain endothelial cells. Addition of oligonucleotides to the NP-Gal actually enhances their internalisation. Moreover, NPs with longer oligonucleotides (40nt) are transported more efficiently than those with shorter oligonucleotides (Fig 3). However, increasing the density of the oligonucleotides on the nanoparticles had a limited effect on the transport rate (Fig 4).

It is known that both size and charge affect the rate of endocytosis of small (<10nm) gold nanoparticles [24]. In general, an increase in charge (positive or negative) promotes internalisation whereas an increase in size reduces it. Attachment of oligonucleotides to the nanoparticles increases the hydrodynamic diameter and adds a large negative charge due to the nucleic acid. The simplest explanation for the increased uptake of the NP-DNA is that the high charge due to the DNA cargo has the major effect on the rate of endocytosis and transcytosis. An increase in the charge of NPs would promote their interaction with the plasma membrane [25]. In addition, the strong negative charge of the NP-DNA could promote their interaction with positively-charged proteins on the endothelial surface. However, in the presence of serum, charged gold nanoparticles bind serum proteins of the opposite charge which effectively flips the charge of the nanoparticle [26]. A positively charged nanoparticle could then interact with the negatively charged endothelial glycocalyx or endothelial surface proteins before internalisation [27, 23]. In addition, the type of bound protein in the corona of the nanoparticles also substantially affects their rate of internalisation [28].

There are two basic mechanisms for transcytosis of nanoparticles—active vesicular endocytosis and transport to the basal membrane, or passive diffusion across apical and basal plasma membranes via the cytosol. This latter mechanism is only available for very small nanoparticles (<10nm) [29]. Vesicular transport is an active and rapid process with vesicles moving on microtubules, whereas cytosolic transfer does not require cellular energy [12]. Our previous studies have shown that gold NPs with a neutral sugar coating move via both cytosolic and vesicular routes, but charged NPs show increased vesicular transcytosis [15]. Since transport was partly inhibited by either chlorpromazine or nystatin, it implied that transport was by clathrin-coated vesicles or caveolae. In this study, the NPs with bound 40nt dsDNA also showed more than 10x higher levels in vesicles than the neutral Gal-NPs while NPs with 20nt dsDNA were intermediate (Fig 3). This result implies that the relationship between charge of the NPs and endocytosis also applies when oligonucleotides are attached. Although the numbers of NP-DNA nanocarriers observed in vesicles and cytosol of the endothelium was similar in the TEM images (Fig 4), this does not mean that equivalent numbers of NPs transfer by each

route. Since the TEM data only provides a single-time point the images cannot show the rate of transcytosis through different compartments, only the numbers of NPs present in each compartment at the defined time-point. In addition, material that has been internalised may be retained or diverted to other intracellular compartments [30, 31] or even be re-internalised [32]. In the case of endothelial cells, transcytosis is a normal function and we have observed NP-DNA being released at the basal plasma membrane from vesicles (Fig 3B). The simplest explanation for these results is that an increase in the charge of the nanoparticles enhances their binding to the apical cell surface and that this can increase movement across the plasma membrane. However at the same time it also greatly increases the rate of NP endocytosis into vesicles. Once inside the cell, NPs are transported within minutes to the basal membrane, while NPs in the cytosol diffuse more slowly. Hence, it is likely that vesicular transport is the major contributor for moving NP-DNA across brain endothelium, with an additional contribution from trans-membrane/cytosolic diffusion.

An important finding was that a substantial amount of the DNA cargo was retained as the NPs cross the endothelium (Fig 5). In oxidising conditions, the covalent linkage of the DNA to the NP is stable for days or weeks. Potentially, release of cargo from the nanoparticles could occur by exchange with glutathione in the cytosol. Experiments done with model nanoparticles indicate that the rate of release depends on the molarity of glutathione and time; exchange occurs over a number of hours at the glutathione concentrations typically found in the cytosol [15]. Since vesicular, transport across the endothelium occurs within 30minutes of endocytosis, the NPs are not exposed to conditions that would release large amounts of the DNA. Moreover, nanoparticles transported by vesicles are not exposed to the reducing conditions found in the cytosol. Nanoparticles may be exposed to low pH in transport vesicles, however these NPs with disulphide-bound ligands are stable in the acidic conditions typically found in vesicles—down to pH 5.0. In support of the argument that the conjugates are stable during endothelial transcytosis: the amount of DNA detected in the endothelium by qPCR was low by comparison with the amount in the basal compartment (Fig 5). Moreover, the DNA detected in the endothelium is likely to be bound to the NPs—i.e. most of the oligonucleotide DNA detected in the endothelium is in transit across the cell.

The high rate of internalisation of the NP-DNA by brain endothelium was reflected in the large numbers of nanoparticles released at the basal membrane. In the transwells, these nanoparticles were trapped between the cells and the filter, but in the hydrogel model they were able to diffuse through the collagen matrix and be taken up by astrocytes (Fig 6). Interestingly, in the astrocytes more nanoparticles were seen in the cytosol than vesicles (S7 Fig in S1 File). This could be advantageous, since oligonucleotide release into cytosol would be required for action of microRNAs, siRNA knockdown or targeting of CRISPR/Cas9 gene-editing.

An important question is whether gold nanocarriers can carry a sufficiently large amount of nucleic acid across the blood-brain barrier in vivo, to be therapeutically useful? The data (Fig 6) indicates up to 5 NP-DNA per $\mu m^3$ in the astrocytes. If we assume that volume of astrocyte is ~3 X$10^2$ $\mu m^3$ per cell [33] and if a single gold nanocarrrier can transport ~4 oligonucleotides (Fig 1), then it would be possible to deliver up to 6000 oligonucleotides per cell. Hence this would be an effective amount to target gene-editing which only requires one targeting oligonucleotide and could be sufficient for short-term modulation of gene expression by Mirs or siRNA interacting with individual mRNAs, which are typically present in the range 10–100 molecules per cell [34].

Another important consideration, is whether these nanocarriers could be used safely in vivo? The size, formulation and capping of gold nanoparticles does affect cytotoxicity for hCMEC/D3 cells in vitro [35] although the nanoparticles used in those studies were larger than in this study and they did not have the C2-galactose cap as used on our nanocarriers.

Other cell types, particularly neurons, could be more sensitive to these 2nm gold NPs than the endothelium, but other studies have shown low toxicity of 5nm negatively-charged gold NPs (carboxylate groups) up to 250μg/ml [36]. Moreover, since the nanocarriers used in our study are capped with C2-galactose and carry a DNA cargo, the gold core is completely shielded by biological molecules. However, the most important argument for in vivo safety of these nanocarriers is that they have been given FDA approval for use with a non-covalently bound insulin cargo and phase 1 and 2 clinical trials have been carried out by Midatech Pharma plc.

This work demonstrates that small gold nanocarriers are effective at transporting 40nt DNA oligonucleotides across brain endothelium and into astrocytes in vitro. It may be possible to further improve transport rates by adding targeting molecules to the nanocarriers, to enhance receptor-mediated transcytosis. However, even the current formulations are capable of transporting therapeutically useful amounts of oligonucleotides to target cells beyond the endothelium.

## Supporting information

**S1 File.**
(DOCX)

**S1 Raw images.**
(PDF)

## Author Contributions

**Conceptualization:** Jane Loughlin, David Male.

**Data curation:** Nayab Fatima.

**Formal analysis:** Nayab Fatima.

**Funding acquisition:** Basil Sharrack, David Male.

**Investigation:** Nayab Fatima, Radka Gromnicova.

**Methodology:** Radka Gromnicova, David Male.

**Project administration:** David Male.

**Supervision:** Radka Gromnicova, Jane Loughlin, David Male.

**Visualization:** Nayab Fatima.

**Writing – original draft:** Nayab Fatima, David Male.

**Writing – review & editing:** Nayab Fatima, Radka Gromnicova, Jane Loughlin, Basil Sharrack, David Male.

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
