## [Decision Letter · Decision Letter 0]

20 Jul 2020

PONE-D-20-21009

Gold nanocarriers for transport of oligonucleotides across brain endothelial cells

PLOS ONE

Dear Dr. Male,

Thank you for submitting your manuscript to PLOS ONE. After careful consideration, we feel that it has merit but does not fully meet PLOS ONE’s publication criteria as it currently stands. Therefore, we invite you to submit a revised version of the manuscript that addresses the points raised during the review process.

We look forward to receiving your revised manuscript.

Kind regards,

Yi Cao

Academic Editor

PLOS ONE

Additional Editor Comments:

Please consider extra experiments to address the issues raised by Review 1. Meanwhile, please address the comments from Reviewer 2, in particular the issue about the side effect of Au NPs. There is almost no comment or discussion about the possible side effects of Au NPs. Although Fig. S5 could indicate little cytotoxic effects of Au NPs, they could still induce some side effects at non-cytotoxic concentrations. Meanwhile, only one cytotoxic assay (Alamar blue assay) may be not enough. Multipe assays are highly recommended.

Reviewers' comments:

Reviewer's Responses to Questions

**Comments to the Author**

1. Is the manuscript technically sound, and do the data support the conclusions?

Reviewer #1: Yes

Reviewer #2: Partly

2. Has the statistical analysis been performed appropriately and rigorously? 

Reviewer #1: Yes

Reviewer #2: Yes

3. Have the authors made all data underlying the findings in their manuscript fully available?

Reviewer #1: Yes

Reviewer #2: Yes

4. Is the manuscript presented in an intelligible fashion and written in standard English?

Reviewer #1: Yes

Reviewer #2: Yes

5. Review Comments to the Author

Reviewer #1: In this manuscript, Nayab et al reported a gold nanocarriers for transport of oligonucleotides across brain endothelial cells. Since brain cells were important for many desease and the paper can be accepted after the following issue were concerned.

1. The mechanism of the gold nanocarriers transport of oligonucleotides should further be enhanced.

2. The SEM images of the gold nanoparticle should be added.

3. Since brain cells were very sensitive, can the author futher discuss the advantanges of this technology?

Reviewer #2: The overall structure is nice to me. But there is a safety concern, what about the binding affinity between the AuNPs and oligonucleotides? Will there be a side effect due to the leakage during the transport process?

6. PLOS authors have the option to publish the peer review history of their article (what does this mean?). If published, this will include your full peer review and any attached files.

Reviewer #1: No

Reviewer #2: No

---

## [Author Response · Author response to Decision Letter 0]

27 Aug 2020

Response to the editorial comment on additional cytotoxicity assays:

This class of NPs with a covalently bound C2- sugar coat and cargo has been approved by the FDA for delivery to humans and phase 1 and 2 clinical trials carried out with an insulin cargo. The FDA submission produced by Midatech Pharma plc included cytotoxicity and pathology studies in rats, mini-pigs, and beagles and genotoxicity in S. typhimurium. For example.

Single dose oral treatment of Wistar rats (12mg/Kg oral) in Wistar rats – ‘No pathological changes were detected in brain, heart, lungs, or spleen’: 

Repeated oral dose toxicity of Wistar rats, up to 1.5mg/Kg over 14 days and followed over 29 days – ‘there were no treatment-related changes in hematology, clinical chemistry, gross pathology or histopathology’.

In genotoxicity tests (micronucleus and comet assay), it has been shown that 5nm gold NPs up to 250μg/ml showed no effect in either assay, provided they were neutral or negatively charged (new ref. 36). We had previously carried out NP cytotoxicity studies on hCMEC/D3 cells with an MTT assay (ref.12)and shown no cytotoxicity at the doses used here.

Additions have been made to the introduction and discussion, to highlight that other assays in vitro and in vivo have been performed with these NPs, albeit with a different cargo

Response to reviewers. 

Reviewer #1

1. The mechanisms of the gold nanocarriers transport of oligonucleotides should be further enhanced.

We have previously investigated the mechanisms of gold nanocarrier transport across hCMEC/D3 cells (ref. 23). The section in the discussion, which cited this study has been expanded and correlated with the new results in this paper. 

2. The SEM images of the gold nanoparticle should be added.

We think the reviewer means transmission EM images, since these nanoparticles are too small to resolve in SEM. An example TEM image of the NPs has now been added as supplementary figure 3 together with a table of core diameters and hydrodynamic diameters for NPs with 1-4 20nt ssDNA oligonucleotides attached. The core size does not change significantly, which implies no aggregation of the NPs when DNA is attached. The hydrodynamic diameter of the NPs is more accurately determined by FPLC as already described in the manuscript and supplementary fig. 3. 

3. Since brain cells were very sensitive, can the author further discuss the advantages of this technology.

Additional discussion and references have been added, which also address the points on safety also raised by the editor. 

Reviewer #2 … But there is a safety concern, what about the binding affinity between the AuNPs and oligonucleotides? Will there be a side effect due to the leakage during the transport process?

Additional notes have been added in the discussion, on the stability of the nanoparticles in the conditions normally found in the cytosol or vesicles. The data indicating that relatively little DNA is detected in the endothelial cells (Fig 6) has been highlighted and discussed.The point is made that oligonucleotides detected in the endothelial cells may still be in transit,(attached to the NPs) and not free DNA.

---

## [Decision Letter · Decision Letter 1]

1 Sep 2020

Gold nanocarriers for transport of oligonucleotides across brain endothelial cells

PONE-D-20-21009R1

Dear Dr. David Male,

We’re pleased to inform you that your manuscript has been judged scientifically suitable for publication and will be formally accepted for publication once it meets all outstanding technical requirements.

Kind regards,

Yi Cao

Academic Editor

PLOS ONE

Additional Editor Comments (optional):

Reviewers' comments:

Reviewer's Responses to Questions

**Comments to the Author**

1. If the authors have adequately addressed your comments raised in a previous round of review and you feel that this manuscript is now acceptable for publication, you may indicate that here to bypass the “Comments to the Author” section, enter your conflict of interest statement in the “Confidential to Editor” section, and submit your "Accept" recommendation.

Reviewer #1: All comments have been addressed

Reviewer #2: (No Response)

2. Is the manuscript technically sound, and do the data support the conclusions?

Reviewer #1: Yes

Reviewer #2: Yes

3. Has the statistical analysis been performed appropriately and rigorously? 

Reviewer #1: Yes

Reviewer #2: Yes

4. Have the authors made all data underlying the findings in their manuscript fully available?

Reviewer #1: Yes

Reviewer #2: Yes

5. Is the manuscript presented in an intelligible fashion and written in standard English?

Reviewer #1: Yes

Reviewer #2: Yes

6. Review Comments to the Author

Reviewer #1: The manuscript has been much improved based on the reviewers' commend. Since it is a very emerging research field and thus the paper can be accepted as it is.

Reviewer #2: It is very nice that you're talking about the stability of the nanoparticles in the cytosol or vesicles.

7. PLOS authors have the option to publish the peer review history of their article (what does this mean?). If published, this will include your full peer review and any attached files.

Reviewer #1: No

Reviewer #2: No

---

## [Editor Report · Acceptance letter]

8 Sep 2020

PONE-D-20-21009R1 

Gold nanocarriers for transport of oligonucleotides across brain endothelial cells 

Dear Dr. Male:

I'm pleased to inform you that your manuscript has been deemed suitable for publication in PLOS ONE. Congratulations! Your manuscript is now with our production department. 

Kind regards, 

on behalf of

Dr. Yi Cao 

Academic Editor

PLOS ONE